# Development and Validation of Nutrition Literacy Questionnaire for Chinese Pre-School Children

**DOI:** 10.3390/nu17101704

**Published:** 2025-05-17

**Authors:** Jing Wen, Xiaoxuan Zhang, Xueqian Yin, Guansheng Ma, Junbo Wang

**Affiliations:** 1Department of Nutrition and Food Hygiene, School of Public Health, Peking University, Beijing 100191, China; wenjing1522@pku.org.cn (J.W.); zxxyingyang@163.com (X.Z.); yinxq@pku.edu.cn (X.Y.); mags@bjmu.edu.cn (G.M.); 2Beijing Key Laboratory of Toxicological Research and Risk Assessment for Food Safety, Peking University, Beijing 100191, China

**Keywords:** pre-school children, nutrition literacy, nutrition literacy questionnaire, instrument assessment

## Abstract

**Background**: This study aims to develop and validate the Nutrition Literacy Questionnaire for Chinese Pre-school Children (NLQ-PSC). **Methods**: The reliability of the questionnaire was determined by internal consistency, the construct validity was assessed by exploratory factor analysis (EFA) and confirmatory factor analysis (CFA), and the content validity was assessed by the Pearson correlation coefficient. In order to analyze the application of NLQ-PSC, we conducted a cross-sectional study among 790 pre-school children. **Results**: From the literature review and qualitative methods, NLQ-PSC was developed, including two dimensions of knowledge and four practice dimensions. The NLQ-PSC questionnaire had good reliability and validity. The average NLQ-PSC score of all participants was 64.1 ± 11.0, and we found that older children, girls, and children who had well-educated parents presented higher nutrition literacy. **Conclusions**: NLQ-PSC has been validated and has shown good reliability and validity, and it could be considered a reliable tool to assess Chinese pre-school children.

## 1. Introduction

Meeting energy and nutritional needs in the early childhood stage is crucial for growth and development, not only for achieving physical and mental potential, but also for achieving socio-economic success in later life [1]. The overall prevalence of obesity in children and adolescents was 8.5% [2], and the global prevalence of deficiency in at least one of three micronutrients (iron, zinc, and vitamin A) was 56% [3]. The Global Burden of Disease Study shows that child and maternal malnutrition was the leading level 2 risk factor for disability-adjusted life years (DALYs) globally in 2019 [4]. Therefore, early adequate nutrition is very important for preventing non-communicable disease later and maintaining lifelong health [5].

Nutrition literacy (NL) is defined as the ability to obtain, process, understand, and use correct nutrition information and nutritional knowledge to make appropriate decisions about food choices [6]. In this sense, being more food and nutrition literate provides one with the necessary aptitude and abilities to help navigate our current food environment [7]. High NL helps people to enhance nutrition food intake and the quality of diet, which have been identified as key components in the promotion and maintenance of healthy dietary practices [7,8]. Childhood is a critical period for the development of eating behaviors and habits that last into adulthood and play a vital role in growth, development, overall health, and the prevention of obesity and other lifelong, diet-related chronic diseases [9,10,11]. However, there were few studies focusing on the nutritional literacy of toddlers or early childhood [7,12].

Assessment tools play an important role in the monitoring and evaluation of nutrition literacy. Several nutritional literacy assessment tools suitable for adults and students have been developed, such as the Nutrition Literacy Assessment Instrument (NLAI) [13], the Food and Nutrition Literacy (FNLIT) [14], the Nutrition Literacy Scale (NLS) [15], the Critical Nutrition Literacy Scale (CNL) [16], the Newest Vital Sign (NVS) [17], and the NLit-IT [18]. In China, nutrition literacy assessment tools were established for school-age children, pregnant women, and adults to the elderly [19,20,21,22]. However, studies related to pre-school children and food/health/nutrition literacy mostly revolved around their caregivers (parents/family caregivers [23] or staff in pre-school childcare institution [24]), and none of these tools can be applied among preschoolers due to the limited cognition, culture, and differences in dietary habits.

Previous studies have focused on the nutritional knowledge and dietary behaviors of pre-school children from both cognitive and behavioral aspects, respectively. Cognitive studies mainly focus on children’s cognition of food and nutrition and the influencing factors, and is often combined with nutrition education intervention programs [25,26,27]. Behavioral research mainly focuses on the formulation and application of scales, and explores the current status and influencing factors of children’s eating behaviors [28,29,30]. Both types of research are relatively one-sided and cannot represent the assessment of nutritional literacy. In addition, the data of the studies among pre-school children are usually collected from the caregivers (parents or teachers), without taking into account that their limited level of cognition also has the ability to accept and finish the survey.

Thus, this study aims to develop the nutrition literacy questionnaire for Chinese pre-school children (NLQ-PSC), while paying attention to children’s cognition of food and nutrition as well as dietary behaviors. We expect to provide an effective tool for assessing and monitoring the nutritional literacy of pre-school children and offer further nutritional education goals.

## 2. Materials and Methods

This study could be divided into 2 parts: the development of the questionnaire and the validation of the questionnaire. It maintained a consistent approach with the previous research of our team on developing assessing tools for nutritional literacy in adults [21], the elderly [22], pregnant women [20], and school-age children [19].

### 2.1. Development of the Nutrition Literacy Questionnaire

#### 2.1.1. Stage 1: The Construction of the Core Components of Nutrition Literacy for Pre-School Children

Based on the literature review, group discussion, and expert consultation, the framework system of nutrition literacy was established. Recommendations related to nutrition and diet for pre-school children were selected from the literature and guidelines about children’s cognition, growth, nutrition, diet, and education, and were formulated as the preliminary core components list of nutrition literacy. Using a two-round Delphi consultation, 9 experts who had more than 10 years of working experience and obtained at least a deputy senior title in the field of nutrition, child and adolescent health, and health education were invited to rate, discuss, and modify the scientific, rationality, applicability, and representativeness of the core components of nutrition literacy for pre-school children via Email.

The detailed elaboration of the methodology and results of the two-round Delphi survey has been published elsewhere [31]. The final core components of the pre-school children’s nutrition literacy were used to set up the questionnaire (Table 1).

#### 2.1.2. Stage 2: The Development of Questionnaire

Taking into consideration the emergence, developing but limited cognitive abilities of pre-school children, and the important role parents played in children’s living and dietary behavior, we assessed their nutrition literacy from both direct (children) and indirect (parent) sides. Based on the core components, we designed the questionnaire to measure nutrition literacy, and for a comprehensive assessment, some components of nutrition literacy corresponded to more than one question. Also, we conducted a two-round Delphi consultation for our questionnaire. Experts who had participated in the formulation of the core components were invited to rate, discuss, and modify the importance and image reasonableness of our questionnaire (1–5 points, 1 = not important, 5 = very important). Finally, a pool of 35 questions was generated, which was mainly composed of children’s and parental sections, including 21 questions for children and 14 questions for parents, and the average score of each question exceeded 3.5 points. In children’s section, accounting for the limited language comprehension and cognitive skills of pre-school children, the questions consisted of picture description questions (“Please identify and verbalize the names of food depicted in the images”), single-choice questions (“Which of these two foods, apples and candy, do you think will lead to weight gain if consumed excessively?”), multiple-choice questions (“Which types of snacks and drinks would you prefer to choose?”), and operation questions (“Please organize the food cards by categorizing them according to food types”).

For the parents’ section, the total questionnaire was composed of basic information and the nutrition literacy of children via an electronic questionnaire. The questions of nutrition literacy included single- and multiple-choice questions (children’s common selection of snacks and beverages) and fill-in-the-blank questions (children’s duration of physical activities), which were related to the living and dietary behavior of children. Single-choice questions related to the frequency and food consumption were rated on a 5-point Likert-type scale, ranging from “not at all” (or “<400 mL”) to “always” (or “≥1000 mL”).

Accounting for the significant developmental changes in cognitive level of children at different ages between 2 and 6 years old, children under 4 could skip the questions beyond their comprehension. According to the evaluation and adjustment suggestions from experts in nutrition education and senior teachers in kindergarten, redundant items were eliminated, and the final questionnaire for children aged 2 to 4 included 35 questions, while 33 questions for children aged 4 to 6. The children’s and parental sections of the questionnaire were given equal weight, and the total questionnaire had a maximum score of 100 points. Based on the rating on each question from experts and professionals, each question was given a different weight.

### 2.2. Validation of the Nutrition Literacy Questionnaire

#### 2.2.1. Data Collection

Children and their parents were recruited via convenience sampling in kindergartens in Beijing, Shandong, Sichuan, and Hunan from November 2019 to October 2021. Eligible children aged 2 to 6 who were able to communicate, understand well, and without severe birth defects, diseases, and one of their parents were enrolled in the survey. Children’s studies were conducted by a single researcher through a face-to-face method, while parents completed the electronic questionnaire released by researchers via smartphone. Based on this, we investigated 1097 children and 1052 parents in total. We also collected information on parents’ education level and annual family income. After combining the data of the children’s questionnaire with the parental section and matching the data according to children’s name, gender, and area, we finally collected 739 child–parent pairs, and the pairing rate was 68.5%.

Before the investigation, we obtained face-to-face verbal confirmation from children and written informed consent was obtained from their parents, and we kept all information confidential. This study was approved by the medical ethics committee of the Peking University Institutional Review Board (approval number: IRB00001052-19120).

#### 2.2.2. Reliability and Validity Tests

The reliability and validity tests were based on the core components of nutrition literacy for pre-school children. To reduce the effects of cognition difference, data from children aged over 4 were used for reliability and validity analysis. Due to the different survey respondents between 2 sections of the questionnaire, we tested the reliability and validity of the questionnaire for children and parents separately. Following the 5–10 times of the amount of questions, 210 children and 140 parents were randomly selected as the sample for the questionnaire reliability and validity analysis.

Cronbach’s alpha coefficient was used to measure the internal consistency, and test–re-test reliability was examined by Spearman–Brown coefficient (split-half analysis). Reliability coefficients of 0.6 or higher were considered acceptable.

Exploratory factor analysis (EFA) and confirmatory factor analysis (CFA) were carried out to explore the construct validity of the questionnaire. The suitability of the data for EFA was performed based on the value of Kaiser–Meyer–Olkin (≥0.6) and Bartlett’s test of sphericity (*p* < 0.05). A maximum variance rotation and principal axis factoring (PAF) were used to explore the existing factorial pattern. CFA was applied to verify the compatibility between the actual measurement of data and the theoretic framework. Chi-squared ratio to degree of freedom (χ^2^/df), root mean square error of approximation (RMSEA), adjusted goodness of fit index (AGFI), goodness of fit index (GFI), parsimony normed fit index (PNFI), and parsimony goodness of fit index (PGFI) were used to evaluate the model. Smaller RMSEA value indicated a better fit: values ≤ 0.05 were considered a good fit (0.05 < RMSEA value ≤ 0.08 was acceptable). Larger AGFI, GFI values (>0.90), PNFI, and PGFI values (>0.50) suggested a good model fit.

### 2.3. Statistical Analysis

Epidata 3.1 software (The Epi Data Association, Odense, Denmark) was used for data entry, and all data received were double-checked. Description information was presented as mean ± standard deviation (SD), numbers (*n*), or proportions (%). Reliability analysis was tested using both Cronbach’s alpha and Spearman–Brown coefficients. The quality of the exploratory factor analysis models was assessed using the KMO, Bartlett’s test of sphericity, and total variance. A multivariate linear regression model was used to investigate factors that affect nutrition literacy. A logistic regression model was performed to analyze the factors affecting the nutrition literacy level of pre-school children. Performed via IBM AMOS V.22.0 software (IBM, Aromonk, NY, USA), confirmatory factor analysis was expressed by χ^2^/df, RMSEA, GFI, AGFI, PCFI, and PNFI. Other statistical analyses were conducted via IBM SPSS V.23.0 software (IBM, Aromonk, NY, USA). Two-sided *p* values < 0.05 were considered statistically significant.

## 3. Results

### 3.1. Demographic Characteristics of Participants

While 52.4% were boys, a total of 739 children aged from 2 to 6 years old, and one of their parents participated in the study, and the statistics were used for the questionnaire application analysis. Among these, 210 child samples (28.4% of the subjects) and 140 parent samples (18.9% of the subjects) were randomly selected for the reliability and validity test of the questionnaire. Table 2 shows a summary of the sociodemographic characteristics of the total study and the reliability and validity test samples.

### 3.2. Reliability and Validity of the Questionnaire

#### 3.2.1. Reliability

For the section of the children’s questionnaire, acceptable reliability was supported by internal consistency (Cronbach’s α was 0.660) and split-half reliability (Spearman–Brown coefficient was 0.729). The Cronbach’s α coefficients for the two dimensions (basic knowledge and living and dietary behaviors) were 0.612 and 0.422, respectively. For the parents’ questionnaire, the Cronbach’s α coefficient was 0.628 and the Spearman–Brown coefficient was 0.642.

#### 3.2.2. Content Validity

In the children’s questionnaire, the Pearson correlation coefficient between two domains (knowledge and understanding, living and dietary behaviors) was 0.306, and the correlation coefficient between each dimension and the overall questionnaire was 0.841 and 0.773 (*p* < 0.05), respectively, which indicated a very strong relationship. The correlation coefficients between each component and the “basic knowledge” domain ranged from 0.340 to 0.654, while the coefficients ranged from 0.251 to 0.549 in the “living and dietary behaviors” domain (the detailed correlation coefficients are shown in Appendix A, Table A1). In the parents’ questionnaire, all of the 14 components belonged to the “living and dietary behaviors” domain, and the correlation coefficients between each component and the “living and dietary behaviors” domain ranged from 0.214 to 0.542 (the detailed correlation coefficients are shown in Appendix A, Table A2). For the entire questionnaire, the Pearson correlation coefficient between knowledge and understanding domain and the living and dietary behaviors domain was 0.292 (*p* < 0.001), and the correlation coefficients between the two dimensions and the total score were 0.766 and 0.839 separately.

#### 3.2.3. Construct Validity

For the children’s questionnaire, an adequate KMO value and significant Bartlett’s test of sphericity (0.692, χ^2^ = 651.701, *p* < 0.001), indicating the questionnaire was suitable for factor analysis. Six factors with eigenvalues greater than 1.0 were extracted, with 49.6% of the total variance explained. Considering the KMO value (0.737) and the significance of Bartlett’s sphericity test (χ^2^ = 366.609, *p* < 0.001), the parents’ questionnaire was also determined to be suitable for factor analysis. Four factors with eigenvalues greater than 1.0 were extracted from the parents’ questionnaire, accounting for 53.211% of the variance.

The validation factor analysis supported the structure of the children’s questionnaire with χ^2^/df = 1.378 (<3.0). The RMSEA was 0.043 (<0.05). The GFI and AGFI values at 0.901 and 0.875 were satisfactory, as both values were close to 0.90 for a good model. The PCFI (0.739) and the PNFI (0.546) values were greater than 0.5. For the parents’ questionnaire, adequate model fit was defined as the χ^2^/df = 1.318 and the RMSEA = 0.048 (<0.08). The GFI, AGFI, PNFI, and PCFI values were 0.914, 0.873, 0.589, and 0.720, respectively, suggesting good construct validity of the questionnaire.

### 3.3. Status of Nutritional Literacy Among Pre-School Children

On the basis that each part of the questionnaire has good reliability and validity, we combined the data of children’s and parents’ questionnaires and calculated the score of nutrition literacy among pre-school children (Table 3). The average score for pre-school children was 64.1 ± 11.0, while the maximum and minimum were 93.2 and 26.7, respectively.

As shown in Table 3, girls presented higher nutrition literacy than boys in living and dietary dimensions (*p* < 0.001). Older children showed higher nutrition literacy, not only in the knowledge and understanding dimension, but also in living and dietary behavior (*p*_all_ < 0.001). Compared to the children who lived in Hunan and Shandong, children in Beijing and Sichuan had higher nutrition literacy, and children in Beijing performed significantly better in both knowledge and behavior dimensions than children in Hunan (*p*_all_ < 0.001). For nutrition literacy, there was no significant difference among different weight status children (*p* > 0.05), but the overweight children had more knowledge and understanding than the normal children (*p* = 0.02). Children with higher-education-level parents showed higher nutrition literacy and performed better in both dimensions. Children with higher average monthly household income, specifically more than CNY 3000 per month, had higher nutrition literacy and performed better in the knowledge and understanding dimension (*p*_all_ < 0.001). Children with more than CNY 5000 household income per month performed better in living and dietary behavior than those whose household income was less than CNY 1000 per month (*p* < 0.001). We also analyzed the differences among six dimensions, which are shown in Appendix A, Table A3.

As shown in Table 4, the Pearson correlation coefficients between the total score and different dimensions ranged from 0.218 to 0.702, showing a strong correlation with the overall questionnaire (*p* < 0.001).

### 3.4. Factors Related to the Nutritional Literacy Among Pre-School Children

Multiple linear regression analysis indicated that age, gender, residence, parents’ education level, and children’s weight status were predictors of nutrition literacy in pre-school children (shown in Table 5). Older children and children who were girls, with higher-education-level parents, overweight would have significantly higher nutrition literacy, while children who lived in lower-income regions would have lower nutrition literacy. Set a score of 80 as the cut-off point for excellent nutrition literacy, logistic regression was also conducted to explore the influencing factors of pre-school children’s nutrition literacy, and results are shown in Table 6. Similarly, excellent scores were more likely to be from older children, girls, and children with highly educated parents. Children who lived in lower-income regions showed lower nutrition literacy. Unlike the multiple linear regression results, compared to children with average monthly household income less than CNY 1000, children with average monthly household income more than CNY 3000 had more excellent nutrition literacy (CNY 3000–5000: OR = 0.003, 95%CI: 0.001–0.787, *p* = 0.03; CNY > 5000: OR = 0.004, 95%CI: 0.002–0.913, *p* = 0.04).

## 4. Discussion

According to our knowledge, this is the first developed Nutrition Literacy Assessment Instrument for pre-school children in China. Through two-round Delphi consultation, NLQ-PSC contains 35 questions, covering 14 key components, grouped into the “knowledge and understanding” and “living and dietary behavior” domains. The structure of NLQ-PSC is similar to other nutrition or food literacy research among children, including nutritional knowledge and functional skills [7]. We found pre-school-FLAT [32] as the only available tool in accessing the food literacy among pre-school children, and differently, FLAT focused more on food and explored the relationship between weight status and food/health; relationship between food quality/quantity and health, and knowing the main food categories; relationship between food and environment; traditional foods and the distribution of foods for breakfast/lunch/dinner. However, apart from cultural differences, nutritional literacy is more comprehensive, which represents the ability to acquire, process, and apply nutritional knowledge, so FLAT is not suitable for the nutritional literacy survey of Chinese preschoolers.

According to Piaget’s theory, pre-school children begin to pay attention to the characteristics of certain aspects of things, and their memory awareness and memory methods continue to improve, and the memory content is more refined [33]. Pre-school children are able to evaluatively categorize foods as healthy or unhealthy [34], and their nutrition knowledge can influence their food choices [11,25]. Unlike other surveys of pre-school children [26,32], our study took into account the cognition ability of pre-school children, and included preschoolers as part of our survey respondents, rather than their parents, guardians, or teachers.

A reliability test measures how consistent or stable the results of the questionnaire are. Due to the different respondents, we tested the reliability and validity of the questionnaire for children and parents separately. A value between 0.6 and 0.7 suggests an acceptable level of reliability [35]. The Cronbach’s α of the overall questionnaire for both children and parents was above 0.6, which indicated our questionnaire had acceptable internal consistency. Due to the limited cognitive ability and attention time of pre-school children, we did not set many questions, which may lead to low internal consistency. Additionally, there are many other possible reasons for a low α value, such as the small sample size, a poor amount of questions in some dimensions, and the content overlap in different dimensions.

Content validity reflects whether the questionnaire items meet the measurement purpose and requirements, while construct validity describes the matching degree between the theoretical hypothesis of the scale and the actual measurement value. Structural equation modeling analyses revealed a good fit to the data (children: χ^2^/df = 1.378, *p* < 0.001, RMSEA = 0.043; parents: χ^2^/df = 1.318, *p* < 0.001, RMSEA = 0.048). These results indicated that we successfully established the structural equation model of NQL-PSC, and the actual measurement basically aligns with the theoretical.

Through NLQ-PSC, we evaluated the nutrition literacy of 790 Chinese pre-school children, whose average score was 64.1 ± 11.0. The full score rate of the knowledge and understanding domain was higher than that of the living and dietary behavior. Looking at the score of each dimension in depth, we found that only half of the preschoolers made the right decision in selecting food and physical activities dimensions, which indicated that preschoolers need to be more guided to make the right food choices and more outdoor activities should be encouraged. However, there was not much difference among each group in the knowledge and understanding domain.

Food preferences and dietary patterns are established during infancy and early childhood, setting the stage for healthy (or unhealthy) eating habits later in life [36,37]. In our study, we found that girls, older children had better nutrition literacy; meanwhile, family demographic factors such as household income level and education level of parents could also influence children’s nutrition literacy level. These results remind us that parental literacy may be related to children’s literacy, which is similar to those of previous studies. Home food environment, parental healthy food intake, and parental food literacy may have a positive effect on preschoolers, improving their [38,39], indicating that the food environment of preschoolers can be improved by improving the nutritional literacy of caregivers. Poor parental literacy may lead to worse health outcomes among children [40]. However, a system review showed that education, interventions, and care to preschoolers may improve child diet quality, but the evidence is very uncertain and likely increases fruit consumption, but healthy eating interventions likely result in little to no difference in consumption of non-core foods and sugar-sweetened beverages [11].

Regrettably, we did not investigate children’s dietary intake, real-world food consumption, or the nutritional literacy of pre-school parents or caregivers in our study. Thus, we the following are our future prospects on the nutrition literacy of Chinese pre-school children: (1) promoting the application of NLQ-PSC, expanding the sample size survey, and conducting dietary intake or food consumption surveys of preschoolers; (2) exploring the association between the nutrition literacy of parents or caregivers and the nutrition literacy, dietary behaviors, nutrition-related knowledge of preschoolers; (3) enriching the types of food used in the survey through the combination with APP or AI technology, and reducing the impact caused by the differences in food culture between northern and southern China.

Moreover, based on the scores of each dimension, our research also provides the directions for nutrition education for pre-school children in the future: (1) strengthening the nutrition education for children in selecting food, by teaching children the nutritional characteristics of various snacks and how to choose snacks; (2) strengthening children’s understanding in the food education, such as the origin, processing, cooking process, and the various forms of food.

There are some advantages in our study. First, to our knowledge, this was the first reported assessment tool to evaluate the nutrition literacy level of preschoolers in China. Second, taking the Chinese Dietary Guidelines, publications of official or professional organizations (WHO, etc.) as the theoretical basis, and consulting with the senior experts from the fields of nutrition, children’s health, and health education, our assessment tool was supported by a solid empirical and theoretical basis. Third, our assessment tool took the cognitive abilities of pre-school children into account and comprehensively assessed their nutritional literacy from both their own frontal survey and their parents’ side survey. This was a breakthrough in the survey methodology of pre-school children, and it obtained good reliability and validity.

Undeniably, the limitations of this study were as follows: (1) The use of a convenience sample for surveying rather than a random sample affected the representativeness of the sample. (2) Some indicators in the process of questionnaire verification may only reach an acceptable level and can be improved in subsequent research. (3) Our study did not explore the relationship between nutritional literacy and overall dietary quality of pre-school children, and did not find a relationship between nutritional literacy and body weight, which may be related to the limited representativeness of the sample size, and should be the main focus of attention in the next step to provide scientific evidence for children’s nutrition education and nutrition improvement programs.

## 5. Conclusions

In conclusion, we developed and validated the NLQ-PSC for Chinese pre-school children. The overall NLQ-PSC is an instrument with satisfactory reliability and validity, which can be used to measure the nutrition literacy of Chinese pre-school children. Undeniably, the sample of this study is limited, and a nationwide survey of nutrition literacy was necessary to identify the target population for further nutrition education to develop targeted interventions to improve nutrition literacy and dietary quality, thus further improving their health.

## Figures and Tables

**Table 1 nutrients-17-01704-t001:** The core components of nutrition literacy for pre-school children.

Domain	Dimension	Component
Knowledge and Understanding	Knowing about food	1. Recognizing common food.
		2. Simply classifying food.
	Knowing about the characteristics of food	3. Knowing about the sources of food.
		4. Knowing about the nutrition characteristics of food.
		5. Simply identifying fresh and sanitary food.
Living and dietary behaviors	Selecting food	6. Light Diet, eating fewer salts, oils, and sugars.
		7. Drinking milk and plenty of water every day and rejecting or consuming fewer sugar-sweetened beverages.
		8. Choosing snacks appropriately, and fruits, milk, and nuts are preferred.
	Eating behaviors	9. Treasuring food and eliminating waste.
		10. No picky eating.
		11. Focus on eating during meal and chewing slowly without distraction and delay.
		12. Eating independently and being able to learn and gradually master the use of tableware.
	Eating safely	13. Washing hands before eating.
	Physical activities	14. Participating in various kinds of physical activities actively and reducing sedentary behaviors.

**Table 2 nutrients-17-01704-t002:** Demographic characteristics of participants, *n* (%).

Characteristics	Total (*n* = 739)	Reliability and Validity Study Among Children (*n* = 210)	Reliability and Validity Study Among Parents (*n* = 140)
Gender			
Boys	387 (52.4)	116 (55.2)	78 (55.7)
Girls	352 (47.6)	94 (44.8)	62 (44.3)
Age (years old)			
2~4	294 (39.8)	-	-
4~6	445 (60.2)	210	140
Residence			
Beijing	230 (31.1)	75 (35.7)	56 (40.0)
Shandong	115 (15.6)	30 (14.3)	23 (16.4)
Sichuan	83 (11.2)	24 (11.4)	19 (13.6)
Hunan	311 (42.1)	81 (38.6)	42 (30.0)
BMI (kg, mean ± SD)	15.9 ± 2.0	15.7 ± 2.1	-
Parents’ education level			
Lower junior college degree	393 (53.2)	105 (50.0)	61 (43.6)
Junior college degree or above	343 (46.4)	104 (49.5)	78 (55.7)
Average monthly household income			
CNY <1000	164 (22.2)	39 (18.6)	23 (16.4)
CNY 1000~3000	172 (23.3)	49 (23.3)	24 (17.1)
CNY 3000~5000	69 (9.3)	16 (7.6)	13 (9.3)
CNY ≥5000	332 (44.9)	106 (50.5)	79 (56.4)

**Table 3 nutrients-17-01704-t003:** Distribution of nutrition literacy among pre-school children (mean ± SD).

Variables (Subjects)	Total Score (100)	Knowledge and Understanding (32.3)	Living and Dietary Behavior (67.7)
**Total (739)**	64.1 ± 11.0	21.7 ± 6.3	42.3 ± 7.4
Gender			
Boys (387)	63.3 ± 10.9	21.8 ± 6.2	41.5 ± 7.6
Girls (352)	64.9 ± 11.0	21.7 ± 6.3	43.2 ± 7.1 *
t	−1.932	0.306	−3.146
*p*	0.054	0.759	0.002
Age (years old)			
2~3 (18)	58.9 ± 11.6 ^ab^	19.3 ± 6.7 ^a^	39.6 ± 7.4 ^a^
3~4 (275)	59.9 ± 10.5 ^ab^	18.6 ± 6.0 ^ab^	41.3 ± 7.0 ^a^
4~5 (300)	63.9 ± 9.8 ^a^	22.1 ± 5.4 ^a^	41.7 ± 7.4 ^a^
5~6 (146)	73.1 ± 8.5	27.2 ± 3.9	45.9 ± 6.9
F	58.673	78.902	15.412
*p*	<0.001	<0.001	<0.001
Residence			
Beijing (230)	70.0 ± 9.6 ^cd^	25.1 ± 4.3 ^d^	44.9 ± 7.7 ^cd^
Shandong (115)	67.6 ± 9.0 ^d^	24.9 ± 5.0 ^d^	42.7 ± 6.8 ^d^
Sichuan (83)	68.3 ± 9.2 ^d^	25.1 ± 4.2 ^d^	43.1 ± 6.7 ^d^
Hunan (311)	57.3 ± 9.3	17.2 ± 5.5	40.1 ± 6.8
F	96.796	270.421	20.948
*p*	<0.001	<0.001	<0.001
Weight status			-
Wasting (13)	63.1 ± 10.7	22.8 ± 6.1	40.2 ± 7.1
Normal (673)	63.9 ± 11.0	21.5 ± 6.3 ^e^	42.4 ± 7.4
Overweight (35)	67.9 ± 10.4	24.7 ± 5.2	43.1 ± 7.3
Obesity (16)	63.6 ± 9.9	23.4 ± 6.6	40.2 ± 6.4
F	1.464	3.428	0.959
*p*	0.223	0.017	0.412
Parent’s education level			
Lower junior college degree (393)	59.4 ± 10.0	19.0 ± 6.1	40.3 ± 7.0
Junior college degree or above (343)	69.4 ± 9.5	24.9 ± 4.9	44.6 ± 7.2
t	−13.963	12.927	−8.075
*p*	<0.001	<0.001	<0.001
Average monthly household income			
CNY <1000 (164)	57.9 ± 9.5 ^gh^	17.1 ± 5.6 ^fgh^	40.8 ± 6.9 ^h^
CNY 1000~3000 (172)	60.1 ± 10.9 ^gh^	19.2 ± 6.3 ^gh^	41.0 ± 7.0
CNY 3000~5000 (69)	65.8 ± 9.9	23.8 ± 5.4	42.0 ± 7.4
CNY ≥5000 (332)	68.8 ± 9.7	24.9 ± 4.6	43.9 ± 7.6
F	55.443	237.302	9.540
*p*	<0.001	<0.001	<0.001

* Compared to boys, the score was different (*p* < 0.05). ^a^ Compared to pre-school children aged 5~6, the score was different (*p* < 0.05). ^b^ Compared to pre-school children aged 4~5, the score was different (*p* < 0.05). ^c^ Compared to pre-school children in Shandong, the score was different (*p* < 0.05). ^d^ Compared to pre-school children in Hunan, the score was different (*p* < 0.05). ^e^ Compared to overweight pre-school children, the score was different (*p* < 0.05). ^f^ Compared to pre-school children with an average monthly household income of CNY 1000~3000, the score was different (*p* < 0.05). ^g^ Compared to pre-school children with an average monthly household income of CNY 3000~5000, the score was different (*p* < 0.05). ^h^ Compared to pre-school children with an average monthly household income of CNY 5000 or above, the score was different (*p* < 0.05).

**Table 4 nutrients-17-01704-t004:** Pearson correlation coefficient among dimensions of NLQ-PSC.

	Total	Knowing About Food	Knowing about the Characteristic of Food	Selecting Food	Eating Behaviors	Eating Safely	Physical Activities
Total		0.643 *	0.630 *	0.641 *	0.702 *	0.370 *	0.218 *
Knowledge and Understanding	0.766 *	0.751 *	0.921 *	0.152 *	0.327 *	0.162 *	0.020
Living and Dietary Behavior	0.839 *	0.312 *	0.203 *	0.772 *	0.729 *	0.413 *	0.308 *

* *p* < 0.001.

**Table 5 nutrients-17-01704-t005:** Multiple linear regression analysis of nutrition-literacy-related factors among Chinese pre-school children.

Variables	B	SE	*β*	t	*p*
Constant	65.84	0.97		67.89	<0.001
Age	4.16	0.43	0.28	9.67	<0.001
Gender (ref: boys)	1.81	0.68	0.08	2.67	0.008
Residence (ref: Beijing)					
Shandong	−2.03	1.0	−0.07	−2.02	0.044
Hunan	−9.76	1.0	−0.44	−9.56	<0.001
Parent’s education level (ref: lower junior college degree)	3.48	0.97	0.16	3.60	<0.001
Weight status (ref: wasting)					
Overweight	3.97	1.60	0.08	2.47	0.014

**Table 6 nutrients-17-01704-t006:** Logistic regression of association between nutrition literacy level and demographic characteristics among pre-school children.

Variables	B	SE	*p*	OR	95% CI	
Constant	−4.28	1.82	0.019	0.014		
Age	0.970	0.207	<0.001	2.639	1.757	3.962
Gender (ref: boys)	0.797	0.320	0.013	2.218	1.184	4.156
Residence (ref: Beijing)						
Hunan	−5.399	1.793	0.003	0.005	0.000	0.196
Parent’s education level (ref: lower junior college degree)	1.026	0.472	0.03	2.789	1.106	7.036
Average monthly household income (ref: CNY < 1000)						
CNY 3000–5000	−3.5	1.664	0.035	0.03	0.001	0.787
CNY > 5000	−3.208	1.59	0.044	0.04	0.002	0.913

## Data Availability

The data presented in this study are available on request from the corresponding author. The data are not publicly available due to privacy.

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
