# Peer review of "Development and Validation of Nutrition Literacy Questionnaire for Chinese Pre-School Children"

_nutrients, 2025, doi:10.3390/nu17101704_

Round 1
Reviewer 1 Report
Comments and Suggestions for Authors
The paper's intended topic is relevant. Human nutrition, particularly for children, poses challenges today. Research in this area could prove controversial, but the authors seem to have found satisfactory solutions.
An important aspect to consider is the English style; the presentation of information must be significantly improved. Another weakness is the limited number of references. This could be addressed by including a literature review section in the manuscript. The results obtained from this research need to be correlated with others, as this approach could significantly enhance the manuscript's value.
The article's similarity percentage (27%) is of considerable importance, especially if it is original research. However, the title, which appears to originate from a reference source, raises doubts.
Author Response
Comment 1: An important aspect to consider is the English style; the presentation of information must be significantly improved.
Response 2: We are very sorry for the mistakes in this manuscript and inconvenience they caused in your reading. The manuscript has been thoroughly revised, polished with the help of editing service and also marked out in the revised manuscript, so we hope it can meet the standard. Thanks so much for your useful comments.
Comment 2: Another weakness is the limited number of references. This could be addressed by including a literature review section in the manuscript. The results obtained from this research need to be correlated with others, as this approach could significantly enhance the manuscript's value.
Response 2: Thank you for your advice. Since similar articles are really limited, it is difficult for us to make a lot of analogies and we can only find Tabacchi’s research as the only available tool in accessing the food literacy among preschool children. In the discussion section, we added the differences between our research and Tabacchi’s research in questionnaire content design and the survey subjects. Meanwhile, we also added the analysis of the children's scores in each dimension and put forward the prospects for the future research on preschoolers’ nutrition literacy in China. The revised version was shown in line 293-313 and line 333-368 in the revised manuscript with red letter and highlights.
Comment 3: The article's similarity percentage (27%) is of considerable importance, especially if it is original research. However, the title, which appears to originate from a reference source, raises doubts.
Response 3: Our team has developed the Nutrition Literacy Questionnaire for Chinese school-age children, pregnant Women, elderly and adults. Based on these studies, we developed and verified the Nutrition Literacy Questionnaire for preschool children. Therefore, there are some similarities in title, method and article structure. We supplemented the description of our team's previous research in Materials and Methods and highlighted the differences, which was shown in line 76-78 with red letter and highlights in the revised manuscript. Thanks for your suggestions!
Reviewer 2 Report
Comments and Suggestions for Authors
This researchers studied the "Development and validation of nutrition literacy questionnaire
for Chinese pre-school children". The study is scientifically sound, interesting and well written. However, I have only a few suggestion.
- regarding the validity of the content/questionnaire, it would be better that authors also consider the validation from few nutrition experts regarding the relevance of the questions.
- discussion section should be improved by comparing the study results with other similar studies conducted in other countries. How this study is different from the other studies, what is the differences that authors observed as compared to the other studies.
- before the conclusion section, authors should add a futuristic approach related with the study results, how this study can be helpful for future nutrition policies for children etc.
Author Response
Comments 1: regarding the validity of the content/questionnaire, it would be better that authors also consider the validation from few nutrition experts regarding the relevance of the questions
Response 1: Thank you for your advice. We also conducted a two-round Delphi consultation about our questionnaire. We are sorry for not mentioning it in our manuscript due to our negligence, we have added the statement of development of questionnaire it in our revised manuscript. The revised version was shown in line 100-106 in the revised manuscript with red letter and highlights.
Comments 2: discussion section should be improved by comparing the study results with other similar studies conducted in other countries. How this study is different from the other studies, what is the differences that authors observed as compared to the other studies.
Response 2: Thank you for your advice. Since similar articles are really limited, it is difficult for us to make a lot of analogies and we can only find Tabacchi’s research as the only available tool in accessing the food literacy among preschool children. In the discussion section, we added the differences between our research and Tabacchi’s research in questionnaire content design and the survey subjects. Meanwhile, we also added the analysis of the children's scores in each dimension and put forward the prospects for the future research on preschoolers’ nutrition literacy in China. The revised version was shown in line 293-313 and line 333-366 in the revised manuscript with red letter and highlights.
Comments 3: before the conclusion section, authors should add a futuristic approach related with the study results, how this study can be helpful for future nutrition policies for children etc.
Response 3: Thank you for your suggestions. We have added prospects and suggestions on nutritional literacy research on preschool children in China and nutritional education related to preschool children in our discussion section. The revised version was shown in line 353-368 with red letter and highlights.
Reviewer 3 Report
Comments and Suggestions for Authors
Dear authors,
The paper is a valuable one since is the first reported development and validation of a nutrition literacy assessment tool for Chinese preschool children — addressing a critical gap in public health research for early childhood.
It has important practical use being a validated tool that can inform public health nutrition education and interventions aimed at preschoolers .
Still, I would suggest to try to asses the actual dietary intake of preschoolers or explore the association between nutrition literacy and real-world food consumption patterns. It would offer the studyy a greater depth.
Author Response
Comments 1: Still, I would suggest to try to assess the actual dietary intake of preschoolers or explore the association between nutrition literacy and real-world food consumption patterns. It would offer the study a greater depth.
Response 1: Thank you for your suggestion. In this study, we focused on the development and validation of our questionnaire, and we regrettably did not investigate children’s dietary intake or food consumption related to the application of the questionnaire. We will conduct these surveys in our future application study. Thank you again for your valuable suggestion!